# Supposedly Equivalent Facts That Aren't? Entity Frequency in Pre-training Induces Asymmetry in LLMs

**Yuan He**[1,2,*,†], **Bailan He**[3,4,*], **Zifeng Ding**[5,*], **Alisia Lupidi**[2,6], **Yuqicheng Zhu**[7,8],
**Shuo Chen**[3], **Caiqi Zhang**[5], **Jiaoyan Chen**[9], **Yunpu Ma**[3,10], **Volker Tresp**[3,10], **Ian Horrocks**[2]
[1]Amazon, [2]University of Oxford, [3]LMU Munich, [4]Siemens AG, [5]University of Cambridge,
[6]Meta, [7]University of Stuttgart, [8]Bosch Center for AI, [9]The University of Manchester,
[10]Munich Center for Machine Learning
lawhy@amazon.com, zd320@cam.ac.uk

## Abstract

Understanding and mitigating hallucinations in Large Language Models (LLMs) is crucial for ensuring reliable content generation. While previous research has primarily focused on "when" LLMs hallucinate, our work explains "why" and directly links model behaviour to the pre-training data that forms their prior knowledge. Specifically, we demonstrate that an asymmetry exists in the recognition of logically equivalent facts, which can be attributed to frequency discrepancies of entities appearing as subjects versus objects. Given that most pre-training datasets are inaccessible, we leverage the fully open-source `OLMo` series by indexing its `Dolma` dataset to estimate entity frequencies. Using relational facts (represented as triples) from `Wikidata5M`, we construct probing datasets to isolate this effect. Our experiments reveal that facts with a high-frequency subject and a low-frequency object are better recognised than their inverse, despite their logical equivalence. The pattern reverses in low-to-high frequency settings, and no statistically significant asymmetry emerges when both entities are high-frequency. These findings highlight the influential role of pre-training data in shaping model predictions and provide insights for inferring the characteristics of pre-training data in closed or partially closed LLMs.[1]

## 1 Introduction

Large Language Models (LLMs) have demonstrated remarkable success in generating fluent and contextually relevant text (Brown et al., 2020; Achiam et al., 2023; Anil et al., 2023; Dubey et al., 2024). However, their tendency to produce hallucinated or factually inconsistent information remains a critical challenge (Huang et al., 2025; Rawte et al., 2023), particularly as these models are increasingly deployed in applications where reliability is paramount (Liu et al., 2023; Huang et al., 2024).

Traditionally, research has focused on identifying the circumstances under which hallucinations occur. For example, Lin et al. (2022) found that LLMs struggle with truthfulness when confronted with conspiracy-style prompts, while Lin et al. (2024) demonstrated their vulnerability to variations in language style through question paraphrasing. Additionally, Berglund et al. (2023) highlighted a structural limitation by showing that LLMs often fail to infer reverse implications correctly when fine-tuned on synthetic forward implications. In contrast to these approaches, our work addresses a more fundamental question: *How does the pre-training data — the very source of an LLM's prior knowledge — influence its propensity to hallucinate?*

---

[*]Equal contribution.
[†]Work done prior to joining Amazon.
[1]See code on GitHub: https://github.com/KRR-Oxford/FactProbe; and datasets on Zenodo: https://doi.org/10.5281/zenodo.15092788.

We posit that one key factor lies in the frequency distribution of entities in the pre-training corpus. As illustrated in Figure 1, an LLM may correctly recognise that the football star Diego Maradona has a sibling named Raul Maradona (a lesser-known individual), yet struggle with the inverse recognition that Raul Maradona has a sibling named Diego Maradona, despite both statements conveying the same fact. Our central hypothesis is that *discrepancies in entity frequencies during pre-training introduce bias into the model's predictive distribution over the correctness of equivalent facts*. We aim to analyse and quantify this phenomenon to gain new insights into how pre-training data influence factual reliability in LLMs.

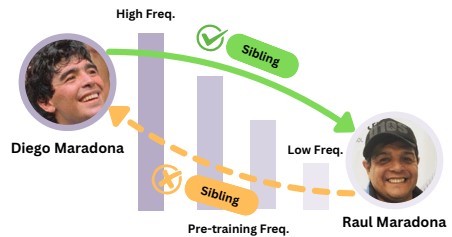

Given that most pre-training datasets are proprietary or otherwise inaccessible (Shi et al., 2023), we leverage the fully open-source OLMo series (Groeneveld et al., 2024; OLMo Team et al., 2024). By indexing its accompanying Dolma pre-training dataset (Soldaini et al., 2024), we are able to estimate entity frequencies reliably. To isolate the effect of entity frequencies, we construct probing datasets from relational facts extracted from Wikidata5M, represented as triples $\langle s, r, o \rangle$. Our experiments reveal a consistent pattern: facts formatted as $\langle s, r, o \rangle$, where the subject $s$ is high-frequency and the object $o$ is low-frequency, are more readily recognised than their logically equivalent inversions, $\langle o, r^{-1}, s \rangle$; when the frequency dynamics

Figure 1: LLMs can exhibit asymmetry when recognising equivalent facts, often identifying facts from high-frequency to low-frequency entities but struggling with the inverse. Shown here is a working example from our tests with the OLMo2-13B model.

are reversed (i.e., low-frequency subject and high-frequency object), the recognition pattern flips; when both entities are high-frequency, the asymmetry is neither prominent nor statistically significant. These findings offer novel insights into an under-explored aspect of model behaviour tied directly to pre-training data, and they motivate further research for inferring characteristics of pre-training corpora – especially for models with undisclosed pre-training datasets.

## 2 Related Work

**Reversal Curse**   The term "reversal curse" was first introduced by Berglund et al. (2023) to describe the structural inability of auto-regressive LLMs to infer "B is entailed by A" when trained on "A entails B". Their work demonstrated this phenomenon using fine-tuned models trained on synthetically crafted datasets, such as names-to-descriptions and questions-to-answers. Further analysis by Zhu et al. (2024) attributed the reversal curse to asymmetries induced by the training dynamics of transformer layers, showing that specific loss functions condition weight updates from one token to another does not necessarily lead the other way round. To mitigate this issue, Golovneva et al. (2024) proposed a reversed training scheme, where models were explicitly fine-tuned with reversed training samples. These works primarily focus on reasoning in a posterior setting, where the model is asked to infer "B is entailed by A" given the premise "A entails B" is learned. In contrast, our work shifts attention to a **priori perspective**, demonstrating that equivalence asymmetry arises due to inherent biases in the pre-training data itself – specifically, from frequency imbalances in subject-object pairs within the model's prior knowledge.

**Probing with Knowledge Bases**   Utilising a knowledge base (KB) to probe LLMs has gained significant interest in recent years, as the structured, high-quality knowledge contained in a KB provides an effective means to verify a model's understanding and mitigate hallucinations. Petroni et al. (2019) first introduced the concept of "LMs-as-KBs" by crafting several probing datasets from different knowledge graphs (KGs) to examine whether an LLM stores relational knowledge. This approach was later extended to more fine-grained KBs, such as temporal KGs (Dhingra et al., 2022) and ontologies (He et al., 2023). In contrast, Zheng et al. (2023) focused on more efficient sampling strategies and alternative metrics to

assess knowledge alignment in LLMs. While these studies primarily quantify how much relational knowledge is stored, our work shifts the focus to probing the asymmetry in fact recognition, demonstrating that LLMs may store equivalent facts in an imbalanced manner.

**Beyond Long-tail Knowledge**  Many studies on long-tail knowledge in LLMs focus on the model's ability to recognise and retrieve facts about infrequent entities (Kandpal et al., 2023; Sun et al., 2024; Li et al., 2024). However, our work extends beyond the traditional long-tail problem by examining cases where subject and object entities exhibit significant discrepancies in pre-training frequency. This difference, rather than overall rarity, leads to systematic inconsistencies in fact recognition. Moreover, while prior works have leveraged Wikidata and other structured knowledge bases to estimate entity frequencies (Wei et al., 2023; Chen et al., 2023; Xin et al., 2024), such sources provide only a partial view of the pre-training corpus, typically accounting for less than 10% of the total training data (Soldaini et al., 2024). In contrast, we derive more accurate frequency estimates directly from the pre-training data itself, ensuring a more representative analysis of the biases in pre-training affecting LLMs (see Appendix D for further discussion).

## 3    Problem Formulation

We define two facts $f_1$ and $f_2$ as equivalent if they express the same underlying statement. Since statements can appear in diverse linguistic forms, we focus on relational facts in KGs to control for linguistic variability and isolate the influence of entity frequencies on model predictions. Below, we formally define *equivalent facts* and *asymmetry in equivalent facts* in the context of KGs, followed by our hypothesis, which we empirically verify through probing experiments.

**Definition 3.1 (Equivalent Facts).**  In a KG, a relational fact is represented as a triple $\langle s, r, o \rangle$, where $s$ is the subject, $r$ is the relation, and $o$ is the object. If $r$ is an *invertible* relation such that $r(s,o) \iff r^{-1}(o,s)$, then $\langle s, r, o \rangle$ is logically equivalent to $\langle o, r^{-1}, s \rangle$.

**Definition 3.2 (Asymmetry in Equivalent Facts).**  This refers to the phenomenon in LLMs where $P(a \mid \langle s, r, o \rangle) \neq P(a \mid \langle o, r^{-1}, s \rangle)$, even though the two triples are logically equivalent. Here, $P(a \mid \cdot)$ indicates the model's *predictive distribution* (Kuhn et al., 2023) over the *correctness* of the fact.

In this context, the correctness label $a$ may be "correct", "incorrect", or "unknown". For our purposes, we focus solely on the $a =$ "correct" case, operating under the assumption that if a model judges a fact as correct, it recognises the fact; otherwise, it does not.

Our hypothesis is that the observed asymmetry originates from significant differences in the frequencies of the subject $s$ and object $o$ in the model's pre-training corpus. Specifically, if $\text{count}(s) \gg \text{count}(o)$, the model is more likely to predict the fact $\langle s, r, o \rangle$ as correct, i.e., $P(a = \text{"correct"} \mid \langle s, r, o \rangle) > P(a = \text{"correct"} \mid \langle o, r^{-1}, s \rangle)$, and conversely, if $\text{count}(o) \gg \text{count}(s)$, $P(a = \text{"correct"} \mid \langle s, r, o \rangle) < P(a = \text{"correct"} \mid \langle o, r^{-1}, s \rangle)$, where $\text{count}(\cdot)$ denotes the frequency of an entity in the pre-training data.

We primarily focus on *symmetric* relations $r$ for which $r^{-1}$ is $r$ itself (e.g., sibling), ensuring that any differences in the model's responses can be more directly attributed to entity frequency effects. By restricting our analysis to these relations, we minimise confounding factors arising from non-symmetric relations (e.g., employedBy vs. employs), which can introduce additional biases due to distinct verbalisations of $r$ and $r^{-1}$.

## 4    Methodology

### 4.1   Indexing Pre-training Corpus

Our objective is to uncover how pre-training biases in LLMs might lead to asymmetry in recognising equivalent facts. Achieving this requires **fully open-source** models (i.e., open

| Relation | # Triples | Question Template | Statement Template |
|----------|-----------|-------------------|--------------------|
| twinTown (P190) | 39,191 | Is $s$ twinned with $o$? | $s$ is twinned with $o$. |
| spouse (P26) | 40,971 | Is $s$ married to $o$? | $s$ is married to $o$. |
| sibling (P3373) | 54,960 | Does $s$ have a sibling named $o$? | $s$ has a sibling named $o$. |
| bordersWith (P47) | 377,967 | Does $s$ border with $o$? | $s$ borders with $o$. |

Table 1: Number of triples and natural language templates for symmetric relations extracted from `Wikidata5M`, where $s$ and $o$ are placeholders for the subject and object, respectively.

weights, training approaches, and datasets), which are rarely available. Among the few meeting these criteria are the `OLMo` series developed by Ai2 (Groeneveld et al., 2024; OLMo Team et al., 2024). They are pre-trained on `Dolma` (Soldaini et al., 2024), a **11TB** open-access corpus,[2] which we index to estimate entity frequencies.

A straightforward approach would be to apply a Named Entity Recognition (NER) model to `Dolma`, but our preliminary trials showed that this is either imprecise (e.g., using a BERT-like NER model) or prohibitively time-consuming (e.g., leveraging a modern decoder-only LLM). Instead, we focus on `Wikidata5M` (Wang et al., 2021), a relatively high-quality subset of `Wikidata` (Vrandečić & Krötzsch, 2014), and perform *string matching* for each of its entities. Because each entity in `Wikidata5M` can have multiple aliases, we simply search the corpus for all possible names of each entity, then sum their occurrences.

However, naive methods such as repeated `grep` commands on large text files become intractable at `Dolma`'s scale. Using data structures like Bloom filters (Marone & Van Durme, 2024) allow fast membership testing but do not offer precise frequency counts, whereas suffix arrays (Nasr et al., 2023) often impose a high memory footprint. To address these limitations, we adopt the FM-index (Ferragina & Manzini, 2000; 2005), a compressed data structure built on the Burrows-Wheeler Transform (BWT) (Burrows, 1994) that facilitates efficient full-text searches over massive corpora. An FM-index typically consists of: *(i)* the BWT of the text, which clusters similar substrings to aid searching; *(ii)* rank and select structures, enabling rapid pattern matching; and *(iii)* occurrence tables, providing precise frequency counts and location information. By applying this indexing scheme, we compress `Dolma` into **4TB** of indexed files and thus achieve reasonably fast lookups for entity frequencies despite the corpus's considerable size.

We also performed a comparative analysis by querying 100 entities using 64 CPUs with both `grep` and the FM-index on `Dolma`. While `grep` took 20 hours due to full-text scans, the FM-index completed the task in just 20 minutes, demonstrating approximately a **60×speedup**. Additional theoretical analysis is provided in Appendix C.

## 4.2 Extracting Relational Facts

We extract relational facts, i.e., triples of the form $\langle s, r, o \rangle$, from `Wikidata5M` and group them by relation $r$.[3] Since our focus is on symmetric relations, we first identify relations that satisfy the symmetric constraint using the Wikidata Query Service (see Appendix B for the SPARQL query). We then filter these relations by retaining only those with more than 10K triples in `Wikidata5M`, resulting in six candidate relations.

However, not all of these relations contain a sufficient number of triples where high-frequency subjects are paired with low-frequency objects (or vice versa for the inverse relation). After further filtering, we select four symmetric relations, each with at least around 1K triples in the extreme high-to-low and low-to-high divisions, i.e., high-frequency entity count $\geq$ 100K and low-frequency entity count $\leq$ 1K. These frequency thresholds are empirical, chosen because it is infeasible to know the complete frequency distribution of all entities in the pre-training corpus. Our analysis shows that among English named

---

[2]https://huggingface.co/datasets/allenai/dolma; version 1.7 was used in our experiments. The file is 4.5TB in its compressed (gzip) form and expands to 11TB when uncompressed.

[3]We focus on entities that have at least one English name; see preprocessing details in Appendix A.

**Question Prompt**

<|system|>
Please evaluate the statement or claim contained in the question. Respond with only one word–either 'Yes' if the claim is correct or 'No' if it is incorrect. Do not include any additional text or commentary.
<|user|>
Does Diego Maradona have a sibling named Raul Maradona?

**Statement Prompt**

<|system|>
Please evaluate the statement or claim. Respond with only one word–either 'True' if the claim is correct or 'False' if it is incorrect. Do not include any additional text or commentary.
<|user|>
Diego Maradona has a sibling named Raul Maradona.

Figure 2: A concrete input example for the triple $\langle \text{DiegoMaradona}, \text{sibling}, \text{RaulMaradona} \rangle$, shown as a *question* (left) and as a *statement* (right). In each case, the system instruction restricts the model's response to a single word: "Yes"/"No" or "True"/"False", respectively.

entities in `Wikidata5M`, approximately two-thirds have frequencies lower than 1K, while only about 5% exceed 100K – a pattern that is consistent with the long-tail distribution typically observed in real-world datasets. Table 1 presents the total number of triples for each selected relation, along with natural language templates for verbalising them (see next section for our probing set-up). The distribution of triples across specific frequency divisions will be reported alongside the results in Section 5.3.

### 4.3 Probing Asymmetry in Equivalent Facts

Following Definition 3.2, we investigate potential asymmetry in an LLM's predictive distribution: $P(a \mid \langle s, r, o \rangle) \neq P(a \mid \langle o, r^{-1}, s \rangle)$, where we designate $\langle s, r, o \rangle$ as the *forward* triple and $\langle o, r^{-1}, s \rangle$ as the *backward* triple, both of which are logically equivalent. Since the model processes forward and backward triples through distinct logits due to its autoregressive nature, comparing these probabilities directly is not straightforward. Instead, we reveal potential asymmetry by comparing how many forward triples versus backward triples are recognised as correct under the same relation $r$.

To assess how an LLM judges the correctness of a relational fact, we frame it as a classification task in which the model needs to produce a single-word answer $a$ given the fact's natural language expression and a task instruction. We employ two prompt templates for each fact: a *question* format and a *statement* format (see Table 1). In the question prompt, the model is asked to respond "Yes" or "No", whereas in the statement prompt, it should answer "True" or "False".[4] A concrete example is provided in Figure 2.

We derive verbalisations for each relation from the corresponding Wikidata descriptions. To better capture the model's potential familiarity with different representations of the same fact, we apply an *inference-time scaling* approach (Snell et al., 2024), introducing variations of entity names for both subject and object. Specifically, we randomly select up to six synonyms from `Wikidata5M` for each entity, resulting in a maximum of 36 prompt variations per fact. Since real-world texts often refer to entities by multiple names, this procedure provides a more robust measure of the model's knowledge. We consider a fact *successfully recognised* if the model produces the correct one-word label ("Yes" for a question or "True" for a statement) in *any* of these variations. We apply the same evaluation criterion to both the forward triple $\langle s, r, o \rangle$ and its backward equivalent $\langle o, r^{-1}, s \rangle$, allowing us to compare recognition accuracies, compute statistical significance, and investigate how entity frequencies affect asymmetry.

**McNemar's Test**   To assess the statistical significance of differences between forward and backward triple recognition, we employ *McNemar's test* (McNemar, 1947), which is designed for comparing paired data. The test statistic is given by $\chi^2 = \frac{(N_{TF} - N_{FT})^2}{N_{TF} + N_{FT}}$, where $N_{TF}$ is the number of paired triples recognised in the forward case but not in the backward case, and $N_{FT}$ is the opposite. Under the null hypothesis $H_0$, we have $N_{TF} = N_{FT}$, indicating no

---

[4]All instruction prompts were generated by `GPT-4o`.

asymmetry. The alternative hypothesis $H_a$ posits $N_{TF} \neq N_{FT}$, indicating asymmetry. The p-value is computed as $1 - F(\chi^2; 1)$, where $F(x; k)$ is the cumulative distribution function (CDF) of the chi-squared distribution with $k$ degree of freedom. A low $p$-value (often $< 0.05$) provides strong evidence to reject $H_0$, showing that the difference in recognition rates is statistically significant and biased in favour of one direction.

## 5 Experiments

### 5.1 Models and Implementations

We focus primarily on the `OLMo` model series due to their direct relevance to the pre-training dataset `Dolma`. In our experiments, we mainly use `OLMo2-32B`[5], the most capable variant at the time of our experiments, and also consider the less capable `OLMo2-13B`[6]. We use the instruction-tuned variants for their ability to follow directions and produce single-word answers, as described in Section 4.3. In addition to the `OLMo` series, we evaluate `Llama3.1-8B`[7] (Dubey et al., 2024) and `Qwen2.5-7B`[8] (Yang et al., 2024), again using their instruction-tuned versions. Although the extent of overlap between their pre-training data and `Dolma` remains unknown, we explore whether the entity frequency information from `Dolma` can be reliable for estimating their pre-training data distributions. For a more accurate and viable probing of the LLM's predictive distribution, we report results with temperature = 0.0 (greedy decoding). Our probing pipeline was implemented based on the `vllm`[9] infrastructure for fast inference (Kwon et al., 2023), and all experiments were conducted on H100 GPUs.

### 5.2 Evaluation Settings

For each relation, we examine three settings: High-to-Low (a high-frequency subject paired with a low-frequency object), Low-to-High (a low-frequency subject paired with a high-frequency object), and High-to-High (both subject and object are high-frequency).[10] In every setting, the forward triples correspond to the original triples from `Wikidata5M`. Although the relation $r$ is symmetric, only one directional instance is typically recorded, meaning that the High-to-Low and Low-to-High settings do not overlap. For the High-to-Low and Low-to-High settings, we fix the high-frequency threshold at 100K and consider three low-frequency ranges: 0-1K, 1K-10K, and 10K-100K. In the High-to-High setting, both the subject and the object exceed the 100K threshold. As described in Section 4.3, we evaluate two prompt templates (question and statement) to examine the effect of prompt phrasing on fact recognition.

### 5.3 Results

**Results for OLMo Models**   We present full results for `OLMo2-32B` in Table 2. Full results for `OLMo2-13B` can be found in Appendix F. Observations for the larger `OLMo2-32B` model are generally consistent with those of `OLMo2-13B`. For each frequency range, we report the total number of triples, the forward and backward triple recognition accuracies, an arrow symbol indicating the preferred direction (⬆ for forward and ⬇ for backward), and the statistical significance, denoted as: $p<0.001$ (***), $p<0.01$ (**), $p<0.05$ (*), or not significant (NS).

Overall, the model demonstrates a clear directional preference: favouring forward triples in the High-to-Low setting and backward triples in the Low-to-High setting. In the most extreme low-frequency range (0-1K), nearly all differences reach top statistical significance (***), with the exception of `bordersWith`. Notably, the spouse relation exhibits the largest

---

[5] https://huggingface.co/allenai/OLMo-2-0325-32B-Instruct
[6] https://huggingface.co/allenai/OLMo-2-1124-13B-Instruct
[7] https://huggingface.co/meta-llama/Llama-3.1-8B-Instruct
[8] https://huggingface.co/Qwen/Qwen2.5-7B-Instruct
[9] https://docs.vllm.ai/

[10]We omit the Low-to-Low setting because the sample size and/or the recognition accuracies in this setting are insufficient to provide statistical meaning.

| Relation | Low Freq. | Total | Question Template | | | | Statement Template | | | |
|---|---|---|---|---|---|---|---|---|---|---|
| | | | Forward | Backward | Diff. | Stat Sig. | Forward | Backward | Diff. | Stat Sig. |
| **High → Low** | | | | | | | | | | |
| twinnedTown | 0-1K | 894 | 0.176 | 0.110 | ↑ | *** | 0.372 | 0.276 | ↑ | *** |
| | 1K-10K | 1667 | 0.219 | 0.113 | ↑ | *** | 0.430 | 0.347 | ↑ | *** |
| | 10K-100K | 3383 | 0.238 | 0.180 | ↑ | *** | 0.487 | 0.469 | ↑ | NS |
| spouse | 0-1K | 1005 | 0.709 | 0.450 | ↑ | *** | 0.647 | 0.383 | ↑ | *** |
| | 1K-10K | 1141 | 0.768 | 0.589 | ↑ | *** | 0.734 | 0.548 | ↑ | *** |
| | 10K-100K | 858 | 0.752 | 0.662 | ↑ | *** | 0.727 | 0.638 | ↑ | *** |
| sibling | 0-1K | 1707 | 0.786 | 0.675 | ↑ | *** | 0.767 | 0.627 | ↑ | *** |
| | 1K-10K | 887 | 0.844 | 0.796 | ↑ | *** | 0.842 | 0.759 | ↑ | *** |
| | 10K-100K | 744 | 0.843 | 0.836 | ↑ | NS | 0.840 | 0.817 | ↑ | NS |
| bordersWith | 0-1K | 12718 | 0.147 | 0.141 | ↑ | NS | 0.141 | 0.135 | ↑ | NS |
| | 1K-10K | 6132 | 0.413 | 0.385 | ↑ | *** | 0.394 | 0.382 | ↑ | NS |
| | 10K-100K | 4397 | 0.507 | 0.485 | ↑ | ** | 0.480 | 0.476 | ↑ | NS |
| **Low → High** | | | | | | | | | | |
| twinnedTown | 0-1K | 934 | 0.095 | 0.171 | ↓ | *** | 0.272 | 0.364 | ↓ | *** |
| | 1K-10K | 1674 | 0.115 | 0.223 | ↓ | *** | 0.364 | 0.444 | ↓ | *** |
| | 10K-100K | 3465 | 0.179 | 0.229 | ↓ | *** | 0.463 | 0.483 | ↓ | * |
| spouse | 0-1K | 1064 | 0.421 | 0.664 | ↓ | *** | 0.374 | 0.605 | ↓ | *** |
| | 1K-10K | 1147 | 0.581 | 0.759 | ↓ | *** | 0.539 | 0.727 | ↓ | *** |
| | 10K-100K | 864 | 0.652 | 0.727 | ↓ | *** | 0.633 | 0.725 | ↓ | *** |
| sibling | 0-1K | 1711 | 0.677 | 0.781 | ↓ | *** | 0.631 | 0.766 | ↓ | *** |
| | 1K-10K | 881 | 0.768 | 0.839 | ↓ | *** | 0.734 | 0.844 | ↓ | *** |
| | 10K-100K | 752 | 0.830 | 0.836 | ↓ | NS | 0.814 | 0.832 | ↓ | NS |
| bordersWith | 0-1K | 13005 | 0.147 | 0.148 | ↓ | NS | 0.140 | 0.143 | ↓ | NS |
| | 1K-10K | 6152 | 0.389 | 0.411 | ↓ | *** | 0.377 | 0.386 | ↓ | NS |
| | 10K-100K | 4418 | 0.488 | 0.507 | ↓ | ** | 0.474 | 0.479 | ↓ | NS |
| **High → High** | | | | | | | | | | |
| twinnedTown | ≥100K | 11103 | 0.231 | 0.232 | ↓ | NS | 0.450 | 0.451 | ↓ | NS |
| spouse | ≥100K | 700 | 0.666 | 0.673 | ↓ | NS | 0.651 | 0.660 | ↓ | NS |
| sibling | ≥100K | 754 | 0.780 | 0.779 | ↑ | NS | 0.776 | 0.782 | ↓ | NS |
| bordersWith | ≥100K | 6254 | 0.674 | 0.676 | ↓ | NS | 0.635 | 0.631 | ↑ | NS |

Table 2: Results of `OLMo2-32B` comparing the statistical differences in recognising forward versus backward relational facts using two template types under High-to-Low, Low-to-High, and High-to-High settings.

forward-backward accuracy differences, ranging from 0.231 to 0.264 across both templates and settings. While this asymmetry persists in higher low-frequency bands (1K-10K and 10K-100K), it tends to diminish as entity frequency increases. For instance, in the High-to-Low setting with the question template, the observed differences for `sibling` are 0.111, 0.048, and 0.007 for the 0-1K, 1K-10K, and 10K-100K ranges, respectively. Non-significant results (NS) typically appear either in the least extreme frequency range (10K-100K) or for relations that are harder to model, such as `bordersWith`.

We also observe that recognition accuracy generally improves as the low-frequency range broadens, aligning with general observations on long-tail knowledge. Among the examined relations, `twinnedTown` and `bordersWith` consistently yield lower accuracies, likely reflecting the model's greater difficulty with geography-related facts (Bhandari et al., 2023).

Finally, in the High-to-High setting where both subject and object entities exceed 100K in frequency, results are consistently non-significant (NS), indicating no clear directional asymmetry when both entities are frequent during pretraining.

**Accuracy Ratios across Models**  Figures 3 and 4 compare the forward/backward recognition accuracy ratios for the spouse and `twinnedTown` relations across four models: `OLMo2-32B`,

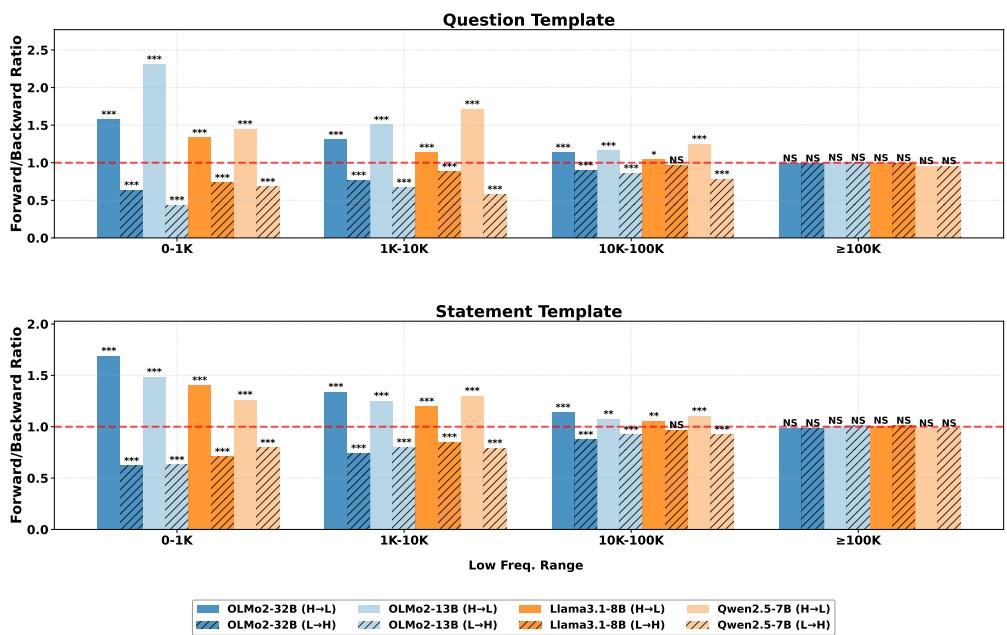

Figure 3: Results of the spouse relation comparing the forward/backward recognition accuracy ratios across models using two template types under High-to-Low, Low-to-High, and High-to-High settings (best viewed in colour).

OLMo2-13B, Llama3.1-8B, and Qwen2.5-7B. These two relations are selected because all models achieve relatively high recognition accuracies, either forward or backward, yielding more stable and meaningful ratios. The red horizontal line at 1.0 denotes parity between forward and backward recognition. For each low-frequency range, both the High-to-Low ratio (plain bar) and the Low-to-High ratio (hatched bar) are presented in the same colour, along with corresponding statistical significance markers above each bar. In the $\geq 100K$ frequency range (i.e., High-to-High), the two bars coincide and thus are identical.

Across both relations, we observe a consistent trend: the High-to-Low setting yields a forward/backward ratio greater than 1, while the Low-to-High setting yields a ratio less than 1. For example, for the spouse relation in the 0-1K low-frequency range, OLMo2-13B with the question template achieves a High-to-Low ratio of approximately 2.3 and a Low-to-High ratio of about 0.4. Notably, even though it is unknown whether Llama3.1-8B and Qwen2.5-7B were pre-trained on Dolma, they exhibit a similar trend with the OLMo models. For instance, both models, using the statement template, attain a High-to-Low ratio above 1.2 and a Low-to-High ratio around 0.8 for the sibling relation in the 0-1K range. In the less extreme low-frequency range (10K-100K), the forward/backward ratios become much less prominent, with almost no difference observed in the High-to-High setting.

Full results for Llama3.1-8B and Qwen2.5-7B are available in Appendix F.

## 6  Discussion

**Rationale Behind the Asymmetry**   We conjecture that the observed asymmetry arises from inherent biases in the pre-training data. In natural language texts, high-frequency entities tend to appear more often as subjects rather than as objects, and subjects typically precede objects in declarative sentences. Consequently, given the autoregressive nature of LLMs, a fact expressed as $\langle s, r, o \rangle$ with a high-frequency entity as $s$ and a low-frequency entity as $o$ is more likely to be recognised than its inverted form $\langle o, r^{-1}, s \rangle$. This tendency in the training data thus leads to the asymmetry defined in Definition 3.2.

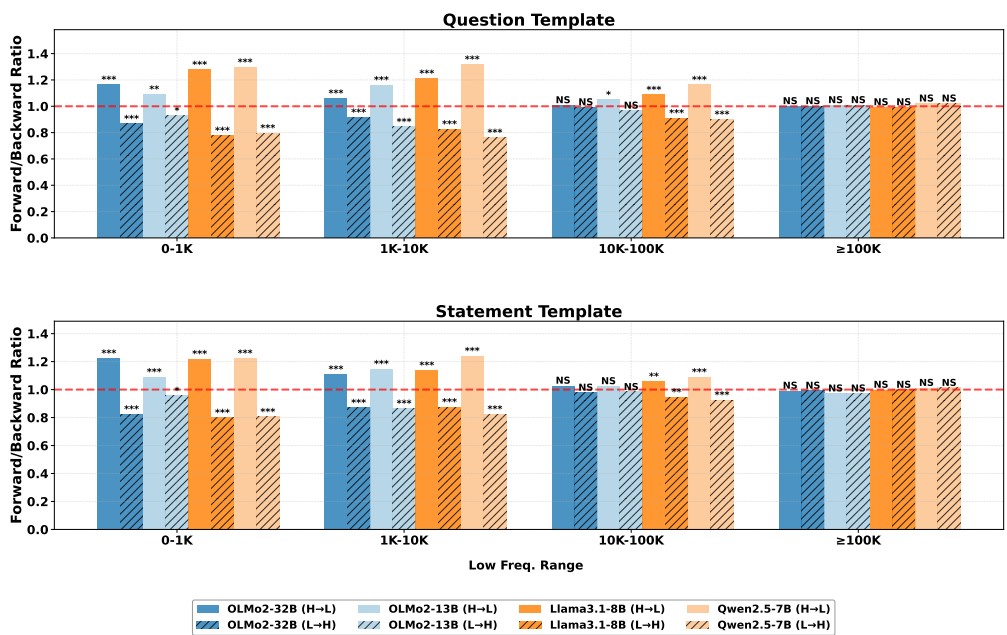

Figure 4: Results of the sibling relation comparing the forward/backward recognition accuracy ratios across models using two template types under High-to-Low, Low-to-High, and High-to-High settings (best viewed in colour).

**Classification Rather than Entity Generation** A common probing setup for similar structural failures in LLMs involves generating free-form answers to a given question (Berglund et al., 2023; Golovneva et al., 2024). In our setting, this would correspond to generating the object given the subject and relation, or generating the subject given the object and the inverse relation. However, this approach essentially examines $P(o \mid \langle s, r, ? \rangle)$ and $P(s \mid \langle o, r^{-1}, ? \rangle)$, which are not expected to be the same unless $r$ is one-to-one. To ensure a rigorous probing of the asymmetry in recognising equivalent facts, we instead formulate the problem as a classification task that aligns with evaluating $P(a \mid \langle s, r, o \rangle)$ and $P(a \mid \langle o, r^{-1}, s \rangle)$, where $\langle s, r, o \rangle$ and $\langle o, r^{-1}, s \rangle$ are logically equivalent.

**Detecting Pre-training Data** Most state-of-the-art LLMs do not publicly disclose detailed information about their pre-training datasets. This lack of transparency has raised ethical and legal concerns, driving efforts to infer the composition of pre-training corpora. Various approaches have been proposed, including prompting models to generate data-specific examples (Sainz et al., 2023), employing statistical methods to detect dataset contamination (Golchin & Surdeanu, 2023; Oren et al., 2023), and using membership inference attacks to determine whether specific data points were present during training (Carlini et al., 2021; Shokri et al., 2017). However, these methods primarily rely on model behaviour, such as output probabilities and loss patterns, which do not necessarily provide direct insight into the structure and distribution of the pre-training data itself. In contrast, our work examines inherent characteristics of pre-training corpora that shape an LLM's factual consistency. Through this, we gain clues about the distribution of entities in closed-source pre-training data. Our results on Llama3.1-8B and Qwen2.5-7B illustrate that these models also exhibit the frequency-based asymmetry, suggesting a possible distribution, or at least the commonality and rarity, of entity mentions in their undisclosed training data.

# 7 Conclusion

In this paper, we investigate the asymmetry in how LLMs recognise logically equivalent facts. Our results indicate that LLMs readily identify relational facts with a high-frequency

subject and a low-frequency object but struggle with the inverse form. By leveraging the fully open-source `OLMo` models and their `Dolma` pre-training data, we accurately estimate entity frequencies and demonstrate a strong correlation between these frequency discrepancies and model performance. Extending our analysis to models with proprietary pre-training data from the `Llama` and `Qwen` families further confirms that the same asymmetry emerges. These findings contribute to understanding how pre-training data characteristics shape model behaviour and motivate future research into mitigating such biases.

## Acknowledgments

This work was supported by Samsung Research UK (SRUK), EPSRC projects UK FIRES (EP/S019111/1) and ConCur (EP/V050869/1), the DAAD programme Konrad Zuse Schools of Excellence in Artificial Intelligence (relAI), the Federal Ministry of Education and Research, the ERC grant AVeriTeC (GA 865958), EU Projects Graph Massivizer (GA 101093202), enRichMyData (GA 101070284) and SMARTY (GA 101140087). The authors thank the International Max Planck Research School for Intelligent Systems (IMPRS-IS) for supporting Yuqicheng Zhu.

## Ethics Statement

Our work probes the asymmetrical behaviour of large language models (LLMs) when presented with logically equivalent facts, aiming to shed light on potential biases arising from entity frequency in training data. To conduct these experiments, we rely on publicly available, open-licensed data (i.e., `Dolma` and `Wikidata5M`) that do not involve collecting or storing personal, private, or sensitive information beyond what is already publicly disclosed. Any real-person examples used (e.g., public figures) appear only to illustrate model behaviour, and do not contain sensitive personal data.

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

## A  Preprocessing of Wikidata5M Entities

Due to the presence of potentially ill-formed entity names in `Wikidata5M`, we apply the following preprocessing steps: *(i)* remove brackets, underscores, and extra spaces; *(ii)* retain only names composed of ASCII characters, digits, dashes, periods, commas, quotation marks, and spaces – components typically found in English named entities; and *(iii)* keep only those entities for which at least one valid name remains after the previous steps.

## B  SPARQL Query for Symmetric Relations

```
SELECT ?property ?propertyLabel
WHERE {
  ?property p:P2302 ?statement .
  ?statement ps:P2302 wd:Q21510862 . # Q21510862 = "symmetric constraint"
  SERVICE wikibase:label {
    bd:serviceParam wikibase:language "[AUTO_LANGUAGE],en"
  }
}
ORDER BY ?propertyLabel
```

Listing 1: SPARQL query that retrieves all Wikidata relations (properties) that have the symmetric constraint.

## C  Efficiency Analysis for Search Algorithms

In our work, we utilise the FM-index for fast entity search in large pre-training corpora, such as `Dolma` (11TB), owing to its significant advantages in both time and space complexity over traditional `grep`-based approaches.

**Time Complexity**   While `grep`, whether employing the Boyer-Moore or Aho-Corasick algorithms, typically requires scanning the entire corpus for each query, resulting in a search time that scales with the corpus size ($\mathcal{O}(N)$), where $N$ represents the total number of characters in the corpus. In contrast, the FM-index performs exact pattern matching in time proportional only to the pattern length ($\mathcal{O}(M)$), where $M$ is the length of the search pattern. When dealing with extremely large datasets like `Dolma`, it becomes practical to partition the FM-index into $K$ separate files. This partitioning allows for parallel processing and more manageable memory requirements during query operations. However, it introduces an additional cost for reporting matches, denoted as $\mathcal{O}(K)$, where $K$ is the number of partitions. In practice, since both $M$ and $K$ are typically much smaller than $N$, the query performance is effectively independent of the corpus size $N$. Constructing the FM-index involves an initial preprocessing cost, which can be as high as $\mathcal{O}(N \log N)$. This preprocessing cost is amortised over numerous queries in a static corpus, leading to substantial overall efficiency gains.

**Space Complexity**   Furthermore, by leveraging the Burrows-Wheeler Transform, the FM-index maintains a compressed representation of the data, thereby significantly reducing the memory footprint compared to `grep`, which must operate on the full, uncompressed text. Partitioning the FM-index into $K$ files also allows for more efficient memory management by loading only the relevant partitions during query execution. This approach ensures that the memory footprint remains manageable, even when working with massive datasets like `Dolma`. This combination of rapid query performance and memory efficiency makes the FM-index a superior choice for our application.

## D  Entity Frequency Analysis

We conduct an additional analysis to highlight the importance of directly estimating entity frequency from pre-training data. As noted in Section 2, some prior studies have identified

long-tail entities based on their rarity in large KGs such as `Wikidata`. To show that this proxy can be inaccurate, we consider all entities in `Wikidata5M` and independently estimate their counts in both the `Dolma` pre-training dataset (see Section 4.1) and `Wikidata` (via the SPARQL query in Listing 2). Because the raw counts are on different scales, we normalise them using the transformation $\log(x + 1)$. We then compute the Pearson linear correlation and Spearman rank correlation between the two sets of normalised counts. The results yield a Pearson correlation of $r = 0.291$ and a Spearman correlation of $\rho = 0.255$, indicating a weak linear relationship between the frequency distributions.

```
1  SELECT (COUNT(DISTINCT ?subject) AS ?subject_count) (COUNT(DISTINCT ?
       object) AS ?object_count) WHERE {{
2    {{
3      ?subject ?p wd:{entity_id} .
4    }} UNION {{
5      wd:{entity_id} ?p ?object .
6    }}
7  }}
```

Listing 2: SPARQL query that retrieves the total count of an entity (appearing as either subject or object) in Wikidata.

## E  Effect of Thinking Before Judging

In the main paper, we focus on prompting models to directly judge the truth of a factual claim (see Figure 2). Here, we explore whether encouraging the model to think step by step (chain-of-thought (CoT) prompting (Wei et al., 2022)) before making a judgment affects its behavior. Specifically, we evaluate `OLMo2-32B` on the spouse relation (Table 3) using a modified instruction: instead of *"Respond with only one word..."*, we prompt the model with *"Let's first think step by step. After reasoning, give the final answer—either 'Yes' if the claim is correct or 'No' if it is incorrect"* (replacing *'Yes'* with *'True'* and *'No'* with *'False'* in the Statement Template). As shown in Table 3, CoT does not alter the observed equivalence asymmetry and it remains consistent across frequency ranges.

| Relation | Low Freq. | Total | Question Template | | | | Statement Template | | | |
|---|---|---|---|---|---|---|---|---|---|---|
| | | | Forward | Backward | Diff. | Stat Sig. | Forward | Backward | Diff. | Stat Sig. |
| **High → Low** | | | | | | | | | | |
| | 0-1K | 1005 | 0.790 | 0.684 | ⬆ | *** | 0.851 | 0.614 | ⬆ | *** |
| spouse | 1K-10K | 1141 | 0.851 | 0.770 | ⬆ | *** | 0.889 | 0.761 | ⬆ | *** |
| | 10K-100K | 858 | 0.829 | 0.803 | ⬆ | NS | 0.893 | 0.830 | ⬆ | *** |
| **Low → High** | | | | | | | | | | |
| | 0-1K | 1064 | 0.648 | 0.747 | ⬇ | *** | 0.613 | 0.831 | ⬇ | *** |
| spouse | 1K-10K | 1147 | 0.762 | 0.838 | ⬇ | *** | 0.753 | 0.881 | ⬇ | *** |
| | 10K-100K | 864 | 0.799 | 0.841 | ⬇ | ** | 0.815 | 0.884 | ⬇ | *** |
| **High → High** | | | | | | | | | | |
| spouse | ≥100K | 700 | 0.803 | 0.793 | ⬆ | NS | 0.829 | 0.831 | ⬇ | NS |

Table 3: Results of `OLMo2-32B` comparing statistical differences in recognising forward and backward relational facts for the *spouse* relation, using the two modified thinking-before-judging template types under High-to-Low, Low-to-High, and High-to-High settings.

## F  Additional Full Results

In the main paper, we report full results for `OLMo2-32B` in Table 2. Here, we amend complete results for `OLMo2-13B` (Table 4), `Llama3.1-8B` (Table 5), and `Qwen2.5-7B` (Table 6).

To see if the observed asymmetry is consistent with even larger models, we conduct further experiments with `Llama3.1-70B`[11] (Table 7). The results show that the asymmetry persists in the most extreme freq range (0-1K), with one exception for bordersWith under the statement template. In less extreme freq ranges, the model either shows the same trend or no statistically significant difference, likely due to its increased capacity.

| Relation | Low Freq. | Total | Question Template | | | | Statement Template | | | |
|---|---|---|---|---|---|---|---|---|---|---|
| | | | Forward | Backward | Diff. | Stat Sig. | Forward | Backward | Diff. | Stat Sig. |
| High → Low | | | | | | | | | | |
| twinnedTown | 0-1K | 894 | 0.032 | 0.015 | ↑ | ** | 0.112 | 0.088 | ↑ | * |
| | 1K-10K | 1667 | 0.042 | 0.018 | ↑ | *** | 0.154 | 0.133 | ↑ | * |
| | 10K-100K | 3383 | 0.076 | 0.047 | ↑ | *** | 0.219 | 0.213 | ↑ | NS |
| spouse | 0-1K | 1005 | 0.337 | 0.146 | ↑ | *** | 0.380 | 0.256 | ↑ | *** |
| | 1K-10K | 1141 | 0.472 | 0.314 | ↑ | *** | 0.616 | 0.492 | ↑ | *** |
| | 10K-100K | 858 | 0.565 | 0.488 | ↑ | *** | 0.681 | 0.634 | ↑ | ** |
| sibling | 0-1K | 1707 | 0.408 | 0.374 | ↑ | ** | 0.583 | 0.535 | ↑ | *** |
| | 1K-10K | 887 | 0.626 | 0.539 | ↑ | *** | 0.745 | 0.649 | ↑ | *** |
| | 10K-100K | 744 | 0.638 | 0.608 | ↑ | * | 0.712 | 0.695 | ↑ | NS |
| bordersWith | 0-1K | 12718 | 0.031 | 0.022 | ↑ | *** | 0.059 | 0.054 | ↑ | * |
| | 1K-10K | 6132 | 0.140 | 0.129 | ↑ | * | 0.211 | 0.186 | ↑ | *** |
| | 10K-100K | 4397 | 0.296 | 0.281 | ↑ | * | 0.371 | 0.357 | ↑ | * |
| Low → High | | | | | | | | | | |
| twinnedTown | 0-1K | 934 | 0.014 | 0.030 | ↓ | ** | 0.092 | 0.116 | ↓ | * |
| | 1K-10K | 1674 | 0.018 | 0.051 | ↓ | *** | 0.136 | 0.156 | ↓ | * |
| | 10K-100K | 3465 | 0.052 | 0.078 | ↓ | *** | 0.211 | 0.225 | ↓ | NS |
| spouse | 0-1K | 1064 | 0.148 | 0.339 | ↓ | *** | 0.243 | 0.387 | ↓ | *** |
| | 1K-10K | 1147 | 0.316 | 0.471 | ↓ | *** | 0.476 | 0.595 | ↓ | *** |
| | 10K-100K | 864 | 0.479 | 0.557 | ↓ | *** | 0.625 | 0.679 | ↓ | *** |
| sibling | 0-1K | 1711 | 0.373 | 0.402 | ↓ | * | 0.538 | 0.563 | ↓ | * |
| | 1K-10K | 881 | 0.524 | 0.621 | ↓ | *** | 0.638 | 0.738 | ↓ | *** |
| | 10K-100K | 752 | 0.616 | 0.637 | ↓ | NS | 0.682 | 0.694 | ↓ | NS |
| bordersWith | 0-1K | 13005 | 0.023 | 0.031 | ↓ | *** | 0.058 | 0.060 | ↓ | NS |
| | 1K-10K | 6152 | 0.123 | 0.136 | ↓ | ** | 0.181 | 0.203 | ↓ | *** |
| | 10K-100K | 4418 | 0.281 | 0.290 | ↓ | NS | 0.354 | 0.366 | ↓ | NS |
| High → High | | | | | | | | | | |
| twinnedTown | ≥100K | 11103 | 0.146 | 0.150 | ↓ | NS | 0.345 | 0.348 | ↓ | NS |
| spouse | ≥100K | 700 | 0.563 | 0.563 | = | NS | 0.647 | 0.641 | ↑ | NS |
| sibling | ≥100K | 754 | 0.569 | 0.564 | ↑ | NS | 0.601 | 0.618 | ↓ | NS |
| bordersWith | ≥100K | 6254 | 0.554 | 0.553 | ↑ | NS | 0.600 | 0.596 | ↑ | NS |

Table 4: Results of `OLMo2-13B` comparing the statistical differences in recognising forward versus backward relational facts using two template types under High-to-Low, Low-to-High, and High-to-High settings.

---

[11] https://huggingface.co/meta-llama/Llama-3.1-70B-Instruct

| Relation | Low Freq. | Total | Question Template | | | | Statement Template | | | |
|---|---|---|---|---|---|---|---|---|---|---|
| | | | Forward | Backward | Diff. | Stat Sig. | Forward | Backward | Diff. | Stat Sig. |
| **High → Low** | | | | | | | | | | |
| twinnedTown | 0-1K | 894 | 0.888 | 0.868 | ↑ | NS | 0.714 | 0.655 | ↑ | *** |
| | 1K-10K | 1667 | 0.901 | 0.878 | ↑ | ** | 0.754 | 0.669 | ↑ | *** |
| | 10K-100K | 3383 | 0.952 | 0.938 | ↑ | ** | 0.791 | 0.741 | ↑ | *** |
| spouse | 0-1K | 1005 | 0.737 | 0.554 | ↑ | *** | 0.623 | 0.444 | ↑ | *** |
| | 1K-10K | 1141 | 0.830 | 0.731 | ↑ | *** | 0.783 | 0.654 | ↑ | *** |
| | 10K-100K | 858 | 0.814 | 0.783 | ↑ | * | 0.763 | 0.723 | ↑ | ** |
| sibling | 0-1K | 1707 | 0.884 | 0.692 | ↑ | *** | 0.813 | 0.667 | ↑ | *** |
| | 1K-10K | 887 | 0.924 | 0.763 | ↑ | *** | 0.868 | 0.763 | ↑ | *** |
| | 10K-100K | 744 | 0.910 | 0.837 | ↑ | *** | 0.835 | 0.789 | ↑ | ** |
| bordersWith | 0-1K | 12718 | 0.412 | 0.460 | ↓ | *** | 0.159 | 0.191 | ↓ | *** |
| | 1K-10K | 6132 | 0.646 | 0.657 | ↓ | NS | 0.330 | 0.311 | ↑ | ** |
| | 10K-100K | 4397 | 0.693 | 0.691 | ↑ | NS | 0.380 | 0.398 | ↓ | * |
| **Low → High** | | | | | | | | | | |
| twinnedTown | 0-1K | 934 | 0.874 | 0.883 | ↓ | NS | 0.654 | 0.713 | ↓ | *** |
| | 1K-10K | 1674 | 0.878 | 0.904 | ↓ | ** | 0.671 | 0.754 | ↓ | *** |
| | 10K-100K | 3465 | 0.943 | 0.952 | ↓ | * | 0.744 | 0.794 | ↓ | *** |
| spouse | 0-1K | 1064 | 0.523 | 0.714 | ↓ | *** | 0.429 | 0.606 | ↓ | *** |
| | 1K-10K | 1147 | 0.712 | 0.810 | ↓ | *** | 0.650 | 0.765 | ↓ | *** |
| | 10K-100K | 864 | 0.786 | 0.814 | ↓ | NS | 0.726 | 0.751 | ↓ | NS |
| sibling | 0-1K | 1711 | 0.687 | 0.884 | ↓ | *** | 0.654 | 0.819 | ↓ | *** |
| | 1K-10K | 881 | 0.765 | 0.926 | ↓ | *** | 0.757 | 0.865 | ↓ | *** |
| | 10K-100K | 752 | 0.832 | 0.914 | ↓ | *** | 0.789 | 0.834 | ↓ | ** |
| bordersWith | 0-1K | 13005 | 0.463 | 0.416 | ↑ | *** | 0.190 | 0.164 | ↑ | *** |
| | 1K-10K | 6152 | 0.655 | 0.645 | ↑ | NS | 0.313 | 0.330 | ↓ | * |
| | 10K-100K | 4418 | 0.691 | 0.693 | ↓ | NS | 0.398 | 0.382 | ↑ | * |
| **High → High** | | | | | | | | | | |
| twinnedTown | ≥100K | 11103 | 0.932 | 0.932 | ↓ | NS | 0.697 | 0.693 | ↑ | NS |
| spouse | ≥100K | 700 | 0.747 | 0.750 | ↓ | NS | 0.681 | 0.674 | ↑ | NS |
| sibling | ≥100K | 754 | 0.871 | 0.877 | ↓ | NS | 0.772 | 0.773 | ↓ | NS |
| bordersWith | ≥100K | 6254 | 0.784 | 0.783 | ↑ | NS | 0.516 | 0.518 | ↓ | NS |

Table 5: Results of `Llama3.1-8B` comparing the statistical differences in recognising forward versus backward relational facts using two template types under High-to-Low, Low-to-High, and High-to-High settings.

| Relation | Low Freq. | Total | Question Template | | | | Statement Template | | | |
|---|---|---|---|---|---|---|---|---|---|---|
| | | | Forward | Backward | Diff. | Stat Sig. | Forward | Backward | Diff. | Stat Sig. |
| **High → Low** | | | | | | | | | | |
| twinnedTown | 0-1K | 894 | 0.393 | 0.375 | ↑ | NS | 0.834 | 0.856 | ↓ | NS |
| | 1K-10K | 1667 | 0.490 | 0.379 | ↑ | *** | 0.916 | 0.878 | ↑ | *** |
| | 10K-100K | 3383 | 0.565 | 0.469 | ↑ | *** | 0.948 | 0.915 | ↑ | *** |
| spouse | 0-1K | 1005 | 0.320 | 0.221 | ↑ | *** | 0.687 | 0.544 | ↑ | *** |
| | 1K-10K | 1141 | 0.428 | 0.250 | ↑ | *** | 0.810 | 0.623 | ↑ | *** |
| | 10K-100K | 858 | 0.486 | 0.389 | ↑ | *** | 0.804 | 0.728 | ↑ | *** |
| sibling | 0-1K | 1707 | 0.582 | 0.449 | ↑ | *** | 0.783 | 0.640 | ↑ | *** |
| | 1K-10K | 887 | 0.667 | 0.506 | ↑ | *** | 0.830 | 0.669 | ↑ | *** |
| | 10K-100K | 744 | 0.726 | 0.624 | ↑ | *** | 0.804 | 0.738 | ↑ | *** |
| bordersWith | 0-1K | 12718 | 0.048 | 0.057 | ↓ | *** | 0.219 | 0.281 | ↓ | *** |
| | 1K-10K | 6132 | 0.168 | 0.171 | ↓ | NS | 0.445 | 0.426 | ↑ | ** |
| | 10K-100K | 4397 | 0.312 | 0.335 | ↓ | *** | 0.555 | 0.565 | ↓ | NS |
| **Low → High** | | | | | | | | | | |
| twinnedTown | 0-1K | 934 | 0.366 | 0.387 | ↓ | NS | 0.857 | 0.833 | ↑ | NS |
| | 1K-10K | 1674 | 0.381 | 0.505 | ↓ | *** | 0.876 | 0.924 | ↓ | *** |
| | 10K-100K | 3465 | 0.468 | 0.568 | ↓ | *** | 0.916 | 0.947 | ↓ | *** |
| spouse | 0-1K | 1064 | 0.209 | 0.308 | ↓ | *** | 0.525 | 0.661 | ↓ | *** |
| | 1K-10K | 1147 | 0.253 | 0.437 | ↓ | *** | 0.626 | 0.791 | ↓ | *** |
| | 10K-100K | 864 | 0.376 | 0.481 | ↓ | *** | 0.728 | 0.788 | ↓ | *** |
| sibling | 0-1K | 1711 | 0.449 | 0.564 | ↓ | *** | 0.634 | 0.788 | ↓ | *** |
| | 1K-10K | 881 | 0.506 | 0.664 | ↓ | *** | 0.670 | 0.813 | ↓ | *** |
| | 10K-100K | 752 | 0.638 | 0.709 | ↓ | *** | 0.743 | 0.803 | ↓ | *** |
| bordersWith | 0-1K | 13005 | 0.061 | 0.047 | ↑ | *** | 0.285 | 0.219 | ↑ | *** |
| | 1K-10K | 6152 | 0.166 | 0.167 | ↓ | NS | 0.424 | 0.441 | ↓ | * |
| | 10K-100K | 4418 | 0.336 | 0.299 | ↑ | *** | 0.563 | 0.552 | ↑ | NS |
| **High → High** | | | | | | | | | | |
| twinnedTown | ≥100K | 11103 | 0.443 | 0.445 | ↓ | NS | 0.869 | 0.869 | ↑ | NS |
| spouse | ≥100K | 700 | 0.501 | 0.524 | ↓ | NS | 0.761 | 0.773 | ↓ | NS |
| sibling | ≥100K | 754 | 0.678 | 0.664 | ↑ | NS | 0.744 | 0.735 | ↑ | NS |
| bordersWith | ≥100K | 6254 | 0.531 | 0.530 | ↑ | NS | 0.688 | 0.688 | ↑ | NS |

Table 6: Results of `Qwen2.5-7B` comparing the statistical differences in recognising forward versus backward relational facts using two template types under High-to-Low, Low-to-High, and High-to-High settings.

| Relation | Low Freq. | Total | Question Template | | | | Statement Template | | | |
|---|---|---|---|---|---|---|---|---|---|---|
| | | | Forward | Backward | Diff. | Stat Sig. | Forward | Backward | Diff. | Stat Sig. |
| High → Low | | | | | | | | | | |
| twinnedTown | 0-1K | 894 | 0.968 | 0.938 | ↑ | ** | 0.971 | 0.932 | ↑ | *** |
| | 1K-10K | 1667 | 0.974 | 0.945 | ↑ | *** | 0.972 | 0.936 | ↑ | *** |
| | 10K-100K | 3383 | 0.982 | 0.947 | ↑ | *** | 0.973 | 0.937 | ↑ | *** |
| spouse | 0-1K | 1005 | 0.902 | 0.834 | ↑ | *** | 0.869 | 0.826 | ↑ | *** |
| | 1K-10K | 1141 | 0.896 | 0.909 | ↓ | NS | 0.897 | 0.909 | ↓ | NS |
| | 10K-100K | 858 | 0.828 | 0.840 | ↓ | NS | 0.831 | 0.850 | ↓ | NS |
| sibling | 0-1K | 1707 | 0.941 | 0.884 | ↑ | *** | 0.948 | 0.902 | ↑ | *** |
| | 1K-10K | 887 | 0.968 | 0.939 | ↑ | *** | 0.964 | 0.950 | ↑ | NS |
| | 10K-100K | 744 | 0.965 | 0.941 | ↑ | ** | 0.958 | 0.950 | ↑ | NS |
| bordersWith | 0-1K | 12718 | 0.919 | 0.908 | ↑ | *** | 0.863 | 0.870 | ↓ | * |
| | 1K-10K | 6132 | 0.953 | 0.939 | ↑ | *** | 0.927 | 0.924 | ↑ | NS |
| | 10K-100K | 4397 | 0.937 | 0.934 | ↑ | NS | 0.917 | 0.913 | ↑ | NS |
| Low → High | | | | | | | | | | |
| twinnedTown | 0-1K | 934 | 0.934 | 0.967 | ↓ | *** | 0.924 | 0.970 | ↓ | *** |
| | 1K-10K | 1674 | 0.939 | 0.973 | ↓ | *** | 0.928 | 0.969 | ↓ | *** |
| | 10K-100K | 3465 | 0.946 | 0.979 | ↓ | *** | 0.933 | 0.973 | ↓ | *** |
| spouse | 0-1K | 1064 | 0.817 | 0.867 | ↓ | *** | 0.809 | 0.837 | ↓ | * |
| | 1K-10K | 1147 | 0.891 | 0.887 | ↑ | NS | 0.887 | 0.875 | ↑ | NS |
| | 10K-100K | 864 | 0.846 | 0.826 | ↑ | NS | 0.852 | 0.829 | ↑ | NS |
| sibling | 0-1K | 1711 | 0.879 | 0.942 | ↓ | *** | 0.897 | 0.938 | ↓ | *** |
| | 1K-10K | 881 | 0.926 | 0.958 | ↓ | *** | 0.941 | 0.961 | ↓ | * |
| | 10K-100K | 752 | 0.940 | 0.955 | ↓ | NS | 0.952 | 0.953 | ↓ | NS |
| bordersWith | 0-1K | 13005 | 0.909 | 0.919 | ↓ | *** | 0.872 | 0.864 | ↑ | ** |
| | 1K-10K | 6152 | 0.938 | 0.950 | ↓ | *** | 0.921 | 0.928 | ↓ | * |
| | 10K-100K | 4418 | 0.932 | 0.936 | ↓ | NS | 0.913 | 0.917 | ↓ | NS |
| High → High | | | | | | | | | | |
| twinnedTown | ≥100K | 11103 | 0.945 | 0.947 | ↓ | NS | 0.923 | 0.926 | ↓ | NS |
| spouse | ≥100K | 700 | 0.791 | 0.806 | ↓ | NS | 0.789 | 0.794 | ↓ | NS |
| sibling | ≥100K | 754 | 0.939 | 0.938 | ↑ | NS | 0.952 | 0.955 | ↓ | NS |
| bordersWith | ≥100K | 6254 | 0.928 | 0.925 | ↑ | NS | 0.903 | 0.904 | ↓ | NS |

Table 7: Results of `Llama3.1-70B` comparing the statistical differences in recognising forward versus backward relational facts using two template types under High-to-Low, Low-to-High, and High-to-High settings.

