# OpenReview forum: "Supposedly Equivalent Facts That Aren’t? Entity Frequency in Pre-training Induces Asymmetry in LLMs"
_colmweb.org/COLM/2025/Conference — COLM 2025_

### Official Review · Reviewer_H939 · 2025-05-11

**Rating:** 4
**Confidence:** 4
**Ethics Flag:** 1

**Summary:**

This paper focuses on a research question of “how do entity frequencies affect the LLM’s knowledge behavior”, and conduct corresponding analyses. To alleviate relation’s side effects (reversal relation expressions), this paper only focuses on asymmetry relations. The authors construct efficient indexing on Dolma with OLMo series models, which increases the research transparency.

**Questions To Authors:**

- How to address the entity linking (or disambiguation) problem? For example, there may be multiple entities with the same name.
- I’m not sure if the filtering process in line 167-168 is reasonable, since the number of KG triples may not represent the number of appearances in the pre-training data.

**Reasons To Accept:**

- This paper validate hypotheses from a data perspective, which is very straightforward and convincing.
- The dataset indexing solution is practical for other pre-training data analyses use cases.

**Reasons To Reject:**

- Since pre-training data are directly related to the model behavior, it is essential to analyze the data. However, only one aspect (frequency) is mentioned in this paper, while others are omitted, such as knowledge correctness, knowledge conflict, multi-lingual problems, etc.
- Although this paper provides empirical analyses about the relationships between training data frequency and reversal curse, it is not clear how these findings would help mitigate such an issue.
- Besides the above aspects, the following research questions may be further addressed:
    - The effects on different entity types (e.g. person, organization, location) and relation types.
    - The effects from entity ambiguities. Do entities with more ambiguities (entities with the same names) lead to hallucinations?
- There are some potential evaluation biases:
    - Why do the evaluation prompts constrain the response to only one word instead of generating the thinking process?
    - There is no background knowledge about the two entities (s and o). How do you ensure the response is 100% correct? Maybe there are different answers under different entities with the same name. For example, we cannot answer “Does Jack have a sibling named Peter?” because there may be a real sibling relationship in the world even they are not famous celebrities.

I may misunderstand some settings or miss some details, please feel free to give me some feedback during the discussion period :)

---

> ### Author Response · Authors · 2025-05-30
>
> We thank the reviewer for the thoughtful and constructive feedback. Below, we address each concern in detail.
>
> ----
>
> *“…only one aspect (frequency) is considered…”*
>
> We focus on frequency as a well-isolated, quantifiable factor hypothesised to have a causal link to equivalence asymmetry. We design **controlled experiments to isolate its effect**. Other aspects like correctness or conflict are important but require different setups. Our dataset is English-only, so multilingual issues are out of scope.
>
> ----
>
> *“...not clear how these findings would mitigate such an issue…”*
>
> While we do not propose a direct mitigation method, our findings provide a diagnostic lens for identifying entity frequency imbalances – even in models with undisclosed training data. Notably, many so-called open-source LLMs release only their model weights, not the underlying data. As a potential mitigation strategy, one could consider **augmenting the training data with reversed subject-object pairs that preserve semantic meaning**, thereby counteracting asymmetries. We will include a brief discussion of this direction in the revised version.
>
> ---
>
> *“...effects on different entity types…”*
>
>
> Our main goal is to isolate the effect of frequency, **independent of entity and relation types**. We design our experiments to demonstrate that the frequency effect generalizes across these categories. While not central to our analysis, our experiments also show that some relation types (e.g., bordersWith) are inherently more difficult than others (e.g., siblings).
>
> ---
>
> *“...entity ambiguities…”*
>
> We mitigate entity ambiguity by including **multiple aliases (synonyms) associated with the same entity** in Wikidata5M during inference (see Section 4.3).
>
> ---
>
> *“...one word instead of generating the thinking process…”*
>
> We constrain responses to one word to enable **exact-match** evaluation, avoiding reliance on LLM-as-judge metrics. This also improves **efficiency,** given the number of tokens needed to be generated.
>
> ---
>
> *“...no background knowledge about the two entities…”*
>
> We mitigate ambiguity by including several known synonyms for each entity (see our earlier response). This helps avoid vague cases like “Mary likes John.” Moreover, when entities appear together, e.g., Mary Lee and Tom Cruise, the context significantly reduces ambiguity. In many cases, the high-frequency entity effectively serves as "background knowledge" for the lower-frequency one.
>
> ---
>
> *“...How to address the entity linking (or disambiguation) problem…”*
>
> Please refer to our earlier response.
>
> ---
>
> *“...filtering process in line 167-168 is reasonable…”*
>
> The filtering in lines 167–168 is used **solely to construct the probing dataset** based on Wikidata5M and is **unrelated to pre-training data**. It ensures sufficient probing examples for statistically meaningful analysis.

---

> > ### Comment · Reviewer_H939 · 2025-05-30
> > **Thanks for your response**
> >
> > > Research scope
> >
> > Thanks for your explanation. I fully understand that you want to present empirical studies on the entity frequencies solely. But I still think the research scope is limited. Besides, I'm curious about what makes entity frequency the most effective factor than the others, although it is intuitive and verified by the reverse curse to some extent. Since this paper does not provide such insights, I'd like to stick to my opinions.
> >
> > > Entity disambiguition
> >
> > I understand that there are less efficient strategies to deal with such data quantities, but this really brings ambiguities. Besides, the context is not trustworthy even with specific entity pairs. You could check studies on distantly supervised relation extraction to see more evidence.
> >
> > > Answer format
> >
> > You can still use exact match in the long CoT mode with a proper instruction. In case of the efficiency, you can sample some cases at each frequency level instead of the whole set.

---

> > ### Author Response · Authors · 2025-06-01
> >
> > We thank the reviewer's prompt reply, below we address further concerns.
> >
> > ---
> >
> > *“...what makes entity frequency the most effective factor than the others…”*
> >
> > We respectfully clarify that our study **does not claim entity frequency is the most effective factor**; rather, we show through controlled experiments that it is an influential factor contributing to equivalence asymmetry. Unlike prior work, we **ground our findings in reliable frequency estimates from pre-training data** and demonstrate consistent results across settings.
> >
> > ---
> >
> > *“...really brings ambiguities...”*
> >
> > We acknowledge that entity ambiguity is a general challenge, but **we make substantial efforts to reduce it by exhaustively including all known aliases from Wikidata5M**. Additionally, **introducing background knowledge, as suggested in the original review, can introduce new sources of bias** and raises nontrivial design questions about what information to include and how to represent it fairly.
> >
> > ---
> >
> > *“...CoT…”*
> >
> > We conducted additional experiments using OLMo2-32B with a CoT prompt on the spouse relation. Due to time constraints during the rebuttal phase and the significant computational cost of generating reasoning traces, we focused on a representative relation to test the robustness of our findings. As shown below, **CoT prompting does not alter the observed equivalence asymmetry and it remains consistent across frequency ranges**.
> >
> > CoT prompt for the Question Template (it is similar for the Statement Template):
> >
> > > Please evaluate the statement or claim contained in the question. Let’s first think step by step. After reasoning, give the final answer—either ‘Yes’ if the claim is correct or ‘No’ if it is incorrect.
> >
> > Results:
> >
> > | Relation | Low Freq. | Total | Question Template | | | | Statement Template | | | |
> > |----------|-----------|-------|--------|--------|------|---------|--------|--------|------|---------|
> > | | | | Forward | Backward | Diff. | Stat Sig. | Forward | Backward | Diff. | Stat Sig. |
> > | **High → Low** | | | | | | | | | | |
> > | spouse | 0-1K | 1005 | 0.790 | 0.684 | ↑ | *** | 0.851 | 0.614 | ↑ | *** |
> > |  | 1K-10K | 1141 | 0.851 | 0.770 | ↑ | *** | 0.889 | 0.761 | ↑ | *** |
> > |  | 10K-100K | 858 | 0.829 | 0.803 | ↑ | NS | 0.893 | 0.830 | ↑ | *** |
> > | **Low → High** | | | | | | | | | | |
> > | spouse | 0-1K | 1064 | 0.648 | 0.747 | ↓ | *** | 0.613 | 0.831 | ↓ | *** |
> > |  | 1K-10K | 1147 | 0.762 | 0.838 | ↓ | *** | 0.753 | 0.881 | ↓ | *** |
> > |  | 10K-100K | 864 | 0.799 | 0.841 | ↓ | ** | 0.815 | 0.884 | ↓ | *** |
> > | **High → High** | | | | | | | | | | |
> > | spouse | ≥100K | 700 | 0.803 | 0.793 | ↑ | NS | 0.829 | 0.831 | ↓ | NS |

---

### Official Review · Reviewer_WBVB · 2025-05-12

**Rating:** 6
**Confidence:** 4
**Ethics Flag:** 1

**Summary:**

This paper investigates a subtle but impactful source of factual inconsistency in large language models (LLMs): asymmetry in recognizing logically equivalent relational facts due to subject-object frequency imbalance during pre-training. The authors use fully open-source models (OLMo-2 series) and their associated Dolma corpus, to estimate entity frequencies with an FM-index and probe models’ ability to recognize facts and their inverse forms. The study confirms LLMs are significantly more likely to recognize high-frequency-to-low-frequency triples than their inverse for symmetric and equivalent relations. Different from previous model-only study, the paper offers a novel and careful empirical analysis from pre-training data distribution, and links it to model factual behavior, with implications for understanding hallucination and knowledge asymmetry in LLMs.

**Questions To Authors:**

1. Address generalization to asymmetric relations in future work — perhaps via a more nuanced probing framework.

2. Broaden discussion on how to mitigate asymmetry — e.g., whether rebalancing subject/object frequencies in fine-tuning could help.

**Reasons To Accept:**

## Empiricism, Data, and Evaluation
- Uses realistic data from Wikidata5M, and precise frequency estimation from 11TB Dolma Pre-training corpus. It probes a wide range of symmetric relations with large triple sets across various frequency bands.

- The evaluation links data to model performance (OLMo) and analyzes other models LLaMA3.1, Qwen2.5). It does some statistical tests (McNemar’s test). The design controls for confounding variables (relation symmetry, verbalization variance) well.


## Technical Depth & Impact
- The authors provide a principled definition of equivalent fact asymmetry (Definition 3.2). Their methodological rigor in isolating subject/object frequency as the main variable is impressive. Meanwhile, the method takes into account the efficiency which naturally fits for the scale of pre-training data.

- Insightfully connects a well-known symptom (“reversal curse”) to a Pre-training data-level root cause.
The framing is not radical, but the idea of “fact asymmetry from data frequency” is relatively novel and opens doors to future directions, e.g. Extension to asymmetric relations; Applications to data attribution, hallucination diagnosis, and LLM alignment auditing.

## Clarity, Honesty, and Trust
The paper is clearly written, with thoughtful diagrams (e.g., forward/backward accuracy ratios).  Care is taken to explain limitations, such as not probing non-symmetric relations. Code and data are open-sourced.

**Reasons To Reject:**

- The work deliberately avoids probing non-symmetric relations, which are arguably more common and interesting in natural language (natural power-low, e.g., “employed by”). The rationale is valid, but it limits generalizability.  The analysis focuses on binary classification which is suitable for symmetric relations, but not fit for asymmetries.

- There is no direct proposal for mitigation — e.g., how one might use this insight to improve pre-training or fine-tuning processes.

---

> ### Author Response · Authors · 2025-06-01
>
> We sincerely thank the reviewer for the detailed and thoughtful feedback. Below, we address each of the raised concerns or questions.
>
> ---
>
> *“...non-symmetric relations…” and Question 1*
>
> We focused on symmetric relations to enable **tightly controlled experiments** isolating subject-object frequency effects. We acknowledge that extending to non-symmetric relations is important. However, the experimental design is non-trivial, as it requires handling **differing prior probabilities and semantics between a relation and its inverse**, and probing while conditioning on these priors. For instance, "employs" and "employedBy" will incur two different logits and thus different priors. We consider this a valuable direction for future work.
>
> ---
>
> *“...proposal for mitigation…” and Question 2*
>
> As a potential mitigation strategy, one could consider **augmenting training data with reversed subject-object pairs that preserve semantic meaning**, thereby counteracting asymmetries. We will include a brief discussion of this direction in the revised version.

---

### Official Review · Reviewer_Ubaa · 2025-05-12

**Rating:** 7
**Confidence:** 4
**Ethics Flag:** 1

**Summary:**

The paper investigates the hypothesis that the "reversal curse" of Berglund et
al.---that an LLM that can correctly answer the question "who was X's
mother?" with "Y" cannot give the answer the symmetrically entailed answer
"X" to the question  "Who was Y's child?"---has its origin in a
well-known bias in LLM QA towards higher frequency entities.

The method is to extract subject-relation-object triples for frequent
symmetrical relations from WikiData5M, then to assess the relative
frequency of  subject and object in the DoLMa pretraining dataset for
the OLMo LLM, and to separate the triples into those with higher
frequency subjects than objects (H -> L) versus the reverse (L -> H),
and the case where all entities are high-frequency (H->H).
It is noted that there is a bias in the DoLMa data towards suject
frequency exceeding object frequency, so presumably the absolute
numbers of triples in the three sets are different, though I couldn't
find those numbers reported.

**Reasons To Accept:**

An interresting property of this selection is that, if LLM QA were
working purely by memorizing the training data, then it is highly
likely that the (for example) the L->H triples are actually attested
in the pretraining data, while their inverses are not.   The extent to
which the LLM performs better on the true but unattested inverses than
on the attested originals is a measure of the strength of the frequency
bias.   As far as I can see, the analysis of which triples are
actually attested in DoLMa has not been done, but it
does make this a very interesting dataset, since it potentially
separates the frequency bias from another well-known factor, the
attestation bias.   I think more could have been made of this point.

For each of the three sets, the success-rate of OLMo (and other
models) in answering a question based on the triple is compared with
that on its symmetrically-entailed inverse.  As predicted by the
frequency bias, H->L questions are more succesfully answered than
their inverses, L->H inverses are more successfully answered than
their originals, and H->H questions are about the same in both
directions.

There is an interesting review of work specifically on the reversal
curse and LLM-as-KB,, including Zue et al and and Petroni.

**Reasons To Reject:**

The motivation for the dataset construction could have been clearer.  In
the end I wasn't sure what the design behind it actually was.

Given the considerable amount of work out there on the frequency bias
in LMs, I think the results are not too surprising:  their significance
lies in the fact that  they are  backed up by actual frequency counts in
the actuual pretraining data for the OLMo LM, making for an
interesting dataset.

---

> ### Author Response · Authors · 2025-06-01
>
> We sincerely thank the reviewer for the detailed and thoughtful feedback. Below, we address each of the raised concerns or questions.
>
> ---
>
> *“… the absolute numbers of triples in the three sets are different, though I couldn't find those numbers reported…”*
>
> The triple counts for each frequency category (H→L, L→H, H→H) are **reported in the “Total” columns of the result tables**. We will clarify this in the text to make it more apparent.
>
> ---
>
> *“...the motivation of the dataset construction…”*
>
> We would like to recall from Introduction (line 31-42) that our motivation is to explain **why** equivalence asymmetry arises by linking it to frequency patterns in the pretraining data, rather than merely stating the effect. To this end, we construct probing datasets with reliable frequency counts drawn directly from the pretraining corpus.
>
> ---
>
> *“...considerable amount of work on the frequency bias in LMs…”*
>
> We appreciate the reviewer’s recognition that our use of actual frequency counts from the pre-training data distinguishes our work. This grounding in pretraining statistics is a key contribution beyond prior studies on frequency bias.

---

### Official Review · Reviewer_HfnW · 2025-05-13

**Rating:** 6
**Confidence:** 4
**Ethics Flag:** 1

**Summary:**

This work examines the influence of entity frequency in LLM pretraining corpora on the ability to recognize factual statements. In particular, they examine symmetric subject, relation, object tuples from the Wikidata knowledge base where the statement is true for either ordering of the subject and object entities (e.g., "A boarders B", "A is the sibling of B"). The authors track effects of each entity's frequency in the Dolma pretraining corpus their ordering in the statement (i.e., whether the more frequent entity is subject or object) on the ability for LMs to recognize whether the statement is true or false. The authors find that LMs are significantly better at assessing the factuality of statements when the more common entity is given as the subject of the statement.

**Questions To Authors:**

Is there a significant reason why treating settings like "high -> low (forward)" and "low -> high (backward)" as distinct settings? The only difference here, as I understand, is how the direction that the relation is stored in WikiData. Is there a common trend in the ordering that Wikidata stores such symmetric relations that would cause us to expect different results in these settings?

**Reasons To Accept:**

This work provides insight into a well known problem in LLM training where models that learn to recognize factual statements with one entity ordering fail to also learn (i.e., models that learn that "A is B" fail to also recognize "B is A").

The work is fairly thorough, testing multiple LMs including the OLMo family of models, which are pretrained on the exact Dolma corpus examined in this work, as well as the Qwen family of models, where pretraining datasets are unknown. The authors also perform their experiments with 2 prompt templates, and find similar results in both settings.

**Reasons To Reject:**

One concern is with the scope of the experiments. While the total number of examples is large, this work analyzes only 4 relations in WikiData. While trends are consistent across these 4 relations, this limited scope casts some concerns regarding the generality of the observed trends. Furthermore, the work only examines LMs up to the 32B scale. Experiments with larger LMs (e.g., API-based Claude, GPT, Gemini) may also yield substantially different trends.

Another potential concern is that the factuality recognition task used in this work is that all tested statements are exclusively true. This lead to results being conflated with LM biases in multiple-choice scenarios. The tested LMs, furthermore, are exclusively Instruciton-tuned models which may have learned refusal / hallucination-prevention behaviors like "be less confident in facts that are about uncommon entities.

---

> ### Author Response · Authors · 2025-06-01
>
> We sincerely thank the reviewer for the detailed and thoughtful feedback. Below, we address each of the raised concerns and provide clarifications or additional results where appropriate.
>
> ---
>
> *“…4 relations in WikiData…”*
>
> We selected the four relations **based on their distribution in Wikidata5M**, ensuring **sufficient sample size** to support robust analysis. We agree that expanding to more relations would strengthen generality and plan to extend our probing datasets using other reliable sources in future work.
>
> ---
>
>  “...Experiments with larger LMs…”
>
> We would like to first note that we already include results from the largest variant of the OLMo models (which directly associate with Dolma). To further address this point, we conducted **additional experiments with LLaMA-3.1-70B-Instruct**, a strong open-weight model. We found that the **asymmetry persists in the most extreme freq range (0–1K)**, with one exception for bordersWith under the statement template. **In less extreme freq ranges, the model either shows the same trend or no statistically significant difference**, likely due to its increased capacity. Please see the result table below:
>
> | Relation | Low Freq. | Total | Question Template | | | | Statement Template | | | |
> |----------|-----------|-------|--------|--------|------|---------|--------|--------|------|---------|
> | | | | Forward | Backward | Diff. | Stat Sig. | Forward | Backward | Diff. | Stat Sig. |
> | **High → Low** | | | | | | | | | | |
> | twinnedTown | 0-1K | 894 | 0.968 | 0.938 | ↑ | ** | 0.971 | 0.932 | ↑ | *** |
> |  | 1K-10K | 1667 | 0.974 | 0.945 | ↑ | *** | 0.972 | 0.936 | ↑ | *** |
> |  | 10K-100K | 3383 | 0.982 | 0.947 | ↑ | *** | 0.973 | 0.937 | ↑ | *** |
> | spouse | 0-1K | 1005 | 0.902 | 0.834 | ↑ | *** | 0.869 | 0.826 | ↑ | *** |
> |  | 1K-10K | 1141 | 0.896 | 0.909 | ↓ | NS | 0.897 | 0.909 | ↓ | NS |
> |  | 10K-100K | 858 | 0.828 | 0.840 | ↓ | NS | 0.831 | 0.850 | ↓ | NS |
> | sibling | 0-1K | 1707 | 0.941 | 0.884 | ↑ | *** | 0.948 | 0.902 | ↑ | *** |
> |  | 1K-10K | 887 | 0.968 | 0.939 | ↑ | *** | 0.964 | 0.950 | ↑ | NS |
> |  | 10K-100K | 744 | 0.965 | 0.941 | ↑ | ** | 0.958 | 0.950 | ↑ | NS |
> | bordersWith | 0-1K | 12718 | 0.919 | 0.908 | ↑ | *** | 0.863 | 0.870 | ↓ | * |
> |  | 1K-10K | 6132 | 0.953 | 0.939 | ↑ | *** | 0.927 | 0.924 | ↑ | NS |
> |  | 10K-100K | 4397 | 0.937 | 0.934 | ↑ | NS | 0.917 | 0.913 | ↑ | NS |
> | **Low → High** | | | | | | | | | | |
> | twinnedTown | 0-1K | 934 | 0.934 | 0.967 | ↓ | *** | 0.924 | 0.970 | ↓ | *** |
> |  | 1K-10K | 1674 | 0.939 | 0.973 | ↓ | *** | 0.928 | 0.969 | ↓ | *** |
> |  | 10K-100K | 3465 | 0.946 | 0.979 | ↓ | *** | 0.933 | 0.973 | ↓ | *** |
> | spouse | 0-1K | 1064 | 0.817 | 0.867 | ↓ | *** | 0.809 | 0.837 | ↓ | * |
> |  | 1K-10K | 1147 | 0.891 | 0.887 | ↑ | NS | 0.887 | 0.875 | ↑ | NS |
> |  | 10K-100K | 864 | 0.846 | 0.826 | ↑ | NS | 0.852 | 0.829 | ↑ | NS |
> | sibling | 0-1K | 1711 | 0.879 | 0.942 | ↓ | *** | 0.897 | 0.938 | ↓ | *** |
> |  | 1K-10K | 881 | 0.926 | 0.958 | ↓ | *** | 0.941 | 0.961 | ↓ | * |
> |  | 10K-100K | 752 | 0.940 | 0.955 | ↓ | NS | 0.952 | 0.953 | ↓ | NS |
> | bordersWith | 0-1K | 13005 | 0.909 | 0.919 | ↓ | *** | 0.872 | 0.864 | ↑ | ** |
> |  | 1K-10K | 6152 | 0.938 | 0.950 | ↓ | *** | 0.921 | 0.928 | ↓ | * |
> |  | 10K-100K | 4418 | 0.932 | 0.936 | ↓ | NS | 0.913 | 0.917 | ↓ | NS |
> | **High → High** | | | | | | | | | | |
> | twinnedTown | ≥100K | 11103 | 0.945 | 0.947 | ↓ | NS | 0.923 | 0.926 | ↓ | NS |
> | spouse | ≥100K | 700 | 0.791 | 0.806 | ↓ | NS | 0.789 | 0.794 | ↓ | NS |
> | sibling | ≥100K | 754 | 0.939 | 0.938 | ↑ | NS | 0.952 | 0.955 | ↓ | NS |
> | bordersWith | ≥100K | 6254 | 0.928 | 0.925 | ↑ | NS | 0.903 | 0.904 | ↓ | NS |
>
> We will include these results in the revised version.
>
> ---
>
> *“...statements are exclusively true…”*
>
> We rely on curated knowledge bases, which contain verified facts. Due to the Open-world Assumption [1], constructing statements that are definitively false is non-trivial -- **absence in the KB does not imply falsehood**. Our setup focuses on isolating frequency and position effects without introducing noise from uncertain labels.
>
> ---
>
> *“...less confident in facts that are about uncommon entities…”*
>
> We would like to note that our setup includes **both high- and low-frequency entities in the same sentence**. Reversing subject and object yields the **same fact** with swapped positions. This isolates the effect of position, not entity presence, effectively **ruling out a simple bias against uncommon entities**.
>
> ---
>
> *“...treating settings like "high -> low (forward)" and "low -> high (backward)" as distinct settings…”*
>
> We retain the original Wikidata ordering to reflect how the relation is typically expressed in real-world usage. This **preserves the natural subject-object structure as curated by human annotators** and avoids introducing artificial symmetry through preprocessing.
>
> ---
>
> [1] Galárraga et al. "AMIE: association rule mining under incomplete evidence in ontological knowledge bases." WWW 2013.

---

### Author Response · Authors · 2025-06-09
**General Response**

We thank all reviewers for their thoughtful feedback and engagement. As the discussion period concludes, we would like to summarise how we have addressed the main concerns:

- **Clarifying misunderstandings**: We responded to points arising from misinterpretations or missing context—for example, the concern about "bias in uncommon entities" (Reviewer HfnW) and "the absence of triple count reports across frequency ranges" (Reviewer Ubaa).

- **Additional experiments**: We conducted new experiments with larger LLMs (LLaMA-3.1-70B, per Reviewer HfnW) and with chain-of-thought (CoT) prompting (per Reviewer H939). In both cases, results remain consistent with our original findings, further reinforcing our conclusions.

- **Reiterating research scope**: We have clarified that our study focuses on how entity frequency in pre-training data induces asymmetry in recognising equivalent facts. Our analysis demonstrates this effect across entity and relation types, and across models, using rigorously controlled experiments (per Reviewer H939). We note that extending the analysis to non-symmetric relations is an important future direction, and have proposed a preliminary mitigation strategy using data augmentation with reversed training examples (per Reviewers WBVB and H939).

---

### Decision · Program_Chairs · 2025-07-08

**Decision:**

Accept

**Comment:**

The paper raises an interesting and well-executed empirical finding: that LLMs show asymmetry in recognizing logically equivalent relational facts depending on entity frequency. But the scope remains narrow—limited to only symmetric relations and frequency as the sole factor. There’s no compelling strategy for mitigation, and the paper stops short of broadening its findings to more realistic, asymmetric scenarios or entity ambiguity issues. While the rebuttal adds helpful clarifications and new results with larger models and CoT prompting, it doesn’t fundamentally address these limitations. Overall, it’s a thoughtful analysis, and while there remains much to be desired, this is a challenging space, and this paper makes a meaningful contribution.